# Coumarins and Hesperetin Inhibit Human Respiratory Syncytial Virus Infection

**DOI:** 10.3390/ijms252413301

**Published:** 2024-12-11

**Authors:** Jéssica Maróstica de Sá, Ilada Thongpan, Jefferson de Souza Busso, Thainá dos Santos Rodrigues, Phylip Chen, Alvaro Luiz Helena, Luis Octavio Regasini, Marcelo Andres Fossey, Ícaro Putinhon Caruso, Fátima Pereira de Souza, Mark Edward Peeples

**Affiliations:** 1Multiuser Center for Biomolecular Innovation, Institute of Biosciences, Letters and Exact Sciences, São Paulo State University (UNESP), São José do Rio Preto 15054-000, SP, Brazil; jessica.marostica@unesp.br (J.M.d.S.); jefferson.busso@unesp.br (J.d.S.B.); thaina.rodrigues@unesp.br (T.d.S.R.); alvaro.helena@unesp.br (A.L.H.); luis.regasini@unesp.br (L.O.R.); marcelo.fossey@unesp.br (M.A.F.); icaro.caruso@unesp.br (Í.P.C.); 2Department of Physics, Institute of Biosciences, Letters and Exact Sciences, São Paulo State University UNESP, São José do Rio Preto 15054-000, SP, Brazil; 3Center for Vaccines and Immunity, Abigail Wexner Research Institute at Nationwide Children’s Hospital, Columbus, OH 43205, USA; ilada.thongpan@nationwidechildrens.org (I.T.); phylip.chen@nationwidechildrens.org (P.C.); mark.peeples@nationwidechildrens.org (M.E.P.); 4Department of Chemical, Institute of Biosciences, Letters and Exact Sciences, São Paulo State University UNESP, São José do Rio Preto 15054-000, SP, Brazil

**Keywords:** RSV, coumarins, hesperetin, HBE culture, A549 cells, HEp-2 cells

## Abstract

Respiratory syncytial virus (RSV) is one of the most prevalent viruses that causes severe acute lower respiratory tract infections (ALRTIs) in the elderly and young children. There is no specific drug to treat RSV, only a broad-spectrum antiviral, ribavirin, which is only used in critical cases. Our research group is investigating antiviral agents of natural origin, such as coumarins and flavonoids, that may help reduce or prevent RSV infection. The cytotoxic concentrations of coumarins and hesperetin were tested on A549 and HEp-2 cells and used in inhibition tests in which 80% of the cells were viable. The anti-RSV action of the molecules was analyzed in A549 and HEp-2 cells and in HBE cell cultures infected with RSV-luc or rgRSV. We also encapsulated the compounds using β-cyclodextrin to improve the permeability and solubility of the molecules. Esculetin and 4-methyl inhibited rgRSV effectively on A549 and HEp-2 cells after 24 hpi, and when they were encapsulated, coumarin, esculetin, and hesperetin presented inhibition against rgRSV in HBE culture. The coumarins inhibit RSV replication in cell culture and even manage to overcome the mucus barriers of the HBE cultures, and β-cyclodextrin was essential for some of the coumarins to enter the cell and therefore to reach their targets.

## 1. Introduction

Respiratory syncytial virus (RSV) is one of the most prevalent viruses that causes severe acute lower respiratory tract infections (ALRTIs) in the elderly and in young children during the first two years of life [1,2]. RSV infection has some complications, such as bronchiolitis, and individuals who have had recurrent infections may present a higher risk of developing asthma that might continue into adulthood [3,4,5]. RSV is a seasonal virus that spreads around the world every year, causing a high number of hospitalizations, with infants under the age of six months being the most vulnerable to severe disease that can result in death, particularly for infants with poor nutrition or in the absence of supportive care [6].

According to a worldwide study in 2015, RSV-ALRI caused around 3.2 million hospital admissions and 59,600 hospital deaths in children under the age of five [7]. However, during the COVID-19 pandemic, an epidemiological survey carried out by the CDC (Center for Disease Control and Prevention) that included 58 counties in 12 USA states showed that from 2022 to 2023, there were approximately four times more hospitalizations for RSV from October to December compared with the same period in 2020–2021. This “rebound” was likely related to the SARS-CoV-2 pandemic since, during this period, there had been social isolation and, consequently, a decrease in RSV infection and therefore natural immunization. The rebound, especially apparent among children, likely reflected a larger group of naïve/susceptible children [8].

At present, the US FDA has licensed two vaccines against RSV, GSK’s Arexvy and Pfizer’s ABRYSVO, for adults over 60 and for pregnant people, respectively. During pregnancy, it is administered between weeks 32 and 36 and provides protection for the newborn up to 6 months of age [9,10]. However, there is no specific drug to treat RSV, only a broad-spectrum antiviral, ribavirin, which is only used in critical cases. Furthermore, there is concern because of the toxicity of this drug, especially for immunocompromised patients, and its inefficient delivery by inhalation [11].

RSV is a single-stranded, negative-sense, enveloped RNA virus in the Mononegavirales order, Pneumoviridae family, and *Orthopneumovirus hominis* species [12]. The RSV genome encodes nine structural and two non-structural proteins [13]. In the process of RSV genome replication and mRNA transcription and translation, the negative strand of the genome is wrapped in the nucleoprotein (N) to form the nucleocapsid. The RNA-dependent RNA polymerase complex (RdRp) is composed of the large polymerase subunit (L), the phosphoprotein cofactor (P) of the polymerase ribonucleoprotein, and the M2-1 cofactor is formed [14,15]. The last residue of the P protein (Phe241) is essential for its binding to the nucleoprotein [16,17], and structural details on the interactions between the N-terminal domain of the nucleoprotein and the C-terminal dipeptide of the P protein (Asp240-Phe241) reported by Ouizougun-Oubari et al. (2015) showed the possibility that this interaction of this hydrophobic pocket (N/P) might be druggable and that the aromatic chemicals (1-benzyl-1Hpyrazole-3,5-dicarboxylate) might bind to it and prevent hRSV replication [18]. Such a possibility might be tested in cell culture experiments. A study carried out by our research group showed that the flavonoid hesperetin (3′,5,7-Trihydroxy-4′-methoxyflavanone) disrupts the interaction between nucleoprotein and phosphoprotein, showing that the phenolic compound binds specifically and selectively to the hydrophobic pocket. Various nuclear magnetic resonance and fluorescence anisotropy analyses corroborated previously published data showing that the hydrophobic pocket of the N-terminal domain of nucleoprotein is a site prone to the binding of small hydrophobic molecules with a chemical structure composed of phenolic rings [19].

Hesperetin is a flavonoid derived from citrus fruits that has gained significant attention for its potential antiviral properties. According to Agrawal et al. (2021), hesperetin is considered a promising candidate for prophylaxis and treatment against SARS-CoV-2 due to its ability to inhibit viral replication. The study utilized in vitro cell culture assays to evaluate the antiviral effects of flavonoids, and these findings show a significant reduction in viral load in treated cells [20]. Khezri and colleagues further elucidate that hesperetin may exert its antiviral effect by modulating the PI3K/AKT signaling pathway, which is crucial for viral entry and replication, indicating a potential role in interfering with SARS-CoV-2 infection. This was demonstrated through molecular docking studies and cellular assays assessing the compound’s impact on key signaling proteins [21]. Additionally, Zalpoor and collaborators propose that hesperetin not only inhibits the virus but also prevents SARS-CoV-2-associated cancer progression by suppressing key intracellular signaling pathways. Their research involved both in vitro and in vivo models, where hesperetin treatment led to a suppression of oncogenic signaling cascades, further supporting its therapeutic potential. These findings highlight the potential of hesperetin as an antiviral agent, capable of modulating various cellular mechanisms to combat both SARS-CoV-2 infection and its associated complications [22].

Given the increasing demand for novel anti-RSV agents and the ongoing research within our group, we are focused on identifying natural antiviral compounds capable of reducing or preventing RSV infection. In this context, the coumarins and flavonoids families represent promising candidates. Coumarins present a benzopyrone heterocyclic in their chemical structure, and they are derived from Tonka beans (*Dipteryx odorata*), from which they were first isolated [23]. They occur naturally in plant roots, seeds, leaves, flowers, peels, and fruits as secondary metabolites [24]. They are of interest in medicinal chemistry due to their various pharmacological potentials and lack of harmful effects on normal cells. These natural molecules of phenolic chemical composition have multiple bioactivities: antibacterial, antiviral, anticancer, antifungal, antifibrotic, anticoagulant, anti-inflammatory, and bronchodilation, among others [25,26,27,28,29,30,31,32,33,34,35,36,37,38,39,40,41,42,43,44,45,46,47,48]. Studies have shown the antiviral action of coumarins and flavonoids on different types of viruses. Yeh and collaborators showed in the plaque reduction assay that angelicin, a furanocoumarin, reduced influenza virus infection by 50% at a concentration of 20 nM and completely at 100 nM [26]. Additionally, Khomenko et al. found antiviral activity from modified coumarin and monoterpenes in HEp-2 cells and some reduction in the RSV titer at all time points [30].

In view of the problem of RSV infections and previous encouraging results from our group, the aim of this study is to analyze the antiviral action of coumarins (coumarin (1,2 Benzopyrone), esculetin (6,7-Dihydroxycoumarin), esculin (6,7-Dihydroxycoumarin-6-glucoside), 4-methyl esculetin (5,7-Dihydroxycoumarin-4-methylcoumarin), umbelliferone (7-Hydroxycoumarin), and hesperetin (3′,5,7-Trihydroxy-4′-methoxyflavanone) flavonoid in cell culture, especially in cultures of well-differentiated primary human bronchial epithelial (HBE) cells.

## 2. Results

### 2.1. Cytotoxicity Assay of Natural and Modified Coumarins and Hesperetin

The results are presented as the percentage of cells surviving exposure to different concentrations of the compounds. In HEp-2 cells, the concentrations of 500 μM for all compounds resulted in cell viability close to 80% (Figure 1), and this was the higher concentration selected for the inhibition assay. When compared with HEp-2 cells, A549 cells demonstrated more sensitivity to the presence of the compounds, since the ideal concentrations at which the cells were 80% viable for coumarin, esculin, umbelliferone, and hesperetin were 300 μM, 125 μM for esculetin, and 75 μM for 4-methyl esculetin (Appendix A).

#### Selectivity Index

Relying on the CC_90_ and CC_100_ values obtained from cytotoxicity assays, we extrapolated the CC_50_ data for each compound. Additionally, IC_50_ values not determined experimentally were estimated through interpolation. These parameters were subsequently utilized to calculate the Selectivity Index (SI) for each compound, providing a measure of therapeutic potential. Table 1 displays the ranking of the molecules.

Among the compounds tested, 4-methyl esculetin exhibited the highest SI of 14.78, with an IC_50_ of 0.07 mM and a CC_50_ of 1.04 mM. This suggests that 4-methyl esculetin is highly effective in inhibiting viral replication while displaying relatively low toxicity to host cells. In contrast, umbelliferone showed the lowest SI of 0.46, which indicates poor selectivity due to its higher IC_50_ (5.14 mM) in relation to its CC_50_ (2.40 mM). This implies that umbelliferone might be more toxic to host cells at the concentrations needed to inhibit the virus, limiting its therapeutic potential. Esculetin and hesperetin displayed moderate SI values of 7.56 and 7.39, respectively. And coumarin and esculin demonstrated lower SI values (2.16 and 1.21, respectively). Overall, 4-methyl esculetin emerges as the most promising compound, however, further in vivo studies and optimization are required to fully establish its clinical potential. Additionally, these compounds may serve as valuable scaffolds for the development of more selective and less toxic antiviral agents.

### 2.2. Inhibitor Test of Natural Coumarins and Hesperetin

An inhibition test was carried out on rgRSV (recombinant green fluorescent protein-expressing RSV, strain A2)-infected HEp-2 cells over the course of 3 days. The concentrations used were those in which 80% of the cells remained viable (coumarin, esculin, and umbelliferone 300 µM, esculetin 150 µM, and 4-methyl 150 µM). Figure 2A shows that esculetin and 4-methyl inhibited rgRSV effectively, while coumarin, esculin, and umbelliferone did not. RSV infection spread from the initially infected cells to other cells by the third day but remains lower in the presence of esculetin and 4-methyl when compared with control and other compounds (Figure 2B).

### 2.3. Quantitative Inhibition of RSV-Luc (Luciferase) Infection and Spread in A549 Cells by Coumarins

The luciferase assays are a powerful tool for monitoring gene expression in cells due to their ultrasensitive detection capacity and wide dynamic range [49]. Using the inhibition test on A549 cells infected with RSV-luc, RSV infection and the inhibitory concentration of each compound were quantified. Figure 3 shows the results in RLU (Relative Lights Units) for each concentration of the compound, exhibiting reduced luciferase gene expression when hesperetin, 4 methyl, and esculetin are present. The inhibition by esculetin and 4-methyl increased with time, while hesperetin inhibition did not (Figure 4). The coumarin, esculin, and umbelliferone did not show inhibition against RSV.

### 2.4. Cytotoxic Assay and Inhibition Test of Modified Umbelliferones on A549 Cells

Two modifications of umbelliferone were made, one added a prenyl group and the other an acetyl group. Appendix A shows results from the MTT test, in which acetylation demonstrated higher cytotoxicity than prenylated umbelliferone. For the inhibition test, concentrations were used following the same criteria for the natural compounds, using the concentrations that resulted in >80% viable cells. The concentrations used were 20 and 300 µM for acetylation and prenylation, respectively. In the inhibition test against RSV-luc, neither of the modified molecules showed inhibition (Figure 5).

### 2.5. Encapsulated Coumarins and Hesperetin on HBE Cultures

As these compounds are all hydrophobic, they may not be able to cross the cell membrane efficiently. Furthermore, the concentration of DMSO needed might be cytotoxic for HBE cultures, the ex vivo model for the natural RSV target cells, and the ciliated airway cells. For these reasons, we encapsulated these compounds using β-cyclodextrin, an oligosaccharide well described in the literature as an excellent carrier for hydrophobic molecules [50,51,52].

We carried out inhibition tests with the encapsulated compounds in HBE cultures, since one of the main interests of our research was to determine if the compounds might be effective in vivo where RSV infects the ciliate cells that are included in these primary cultures. HBE are primary cells differentiated from human airway basal cells and are important cell models for investigating respiratory viruses in the cells that they infect in vivo [53]. First, we checked the cytotoxicity of the encapsulated compounds in A549 cells (Appendix A), and β-cyclodextrin presented lower cytotoxicity, only decreasing about 5% of the available cells. In the inhibitor experiment in HBE cultures, coumarin, esculetin, and hesperetin caused a decrease in RSV infection of approximately 25%, 15%, and 7%, respectively (Figure 6). In the inhibition assays on A549 cells, the coumarin, when not encapsulated, showed no reduction in infection, and hesperetin, although considerably low, when encapsulated, showed inhibition at 48 hpi, which was not seen in the tests in which it was not encapsulated.

### 2.6. Comparative Evaluation of Antiviral Effects

In the statistical tests, most compounds did not show statistically significant differences in viral inhibition in HEp-2 cells (Figure 7A). Specifically, coumarin (*p* = 0.1869), esculetin (*p* = 0.1123), esculin (*p* = 0.2098), and umbelliferone (*p* = 0.4078) did not produce significant inhibition effects, as their adjusted *p*-values were above the significance threshold (*p* > 0.05), indicating no significant differences. The source of variation analysis further supported these findings. For the timer variable, the *p*-value of 0.0096 was below the significance threshold (*p* < 0.05), indicating that the timer factor significantly contributed to the total variation (41.79% of total variation). However, the treatment variable did not show a significant effect on viral inhibition, with a *p*-value of 0.1273 (*p* > 0.05), meaning the treatment had no significant impact on the total variation (30.90% of total variation). Nevertheless, 4-methyl showed a statistically significant difference (*p* = 0.0397), with an adjusted *p*-value below the threshold (*p* < 0.05), suggesting that 4-methyl has a significant viral inhibition compared with the positive control.

In A549 cells, the multiple comparisons test (Figure 7B) exhibited that coumarin, esculetin, esculin, 4-methyl, and umbelliferone demonstrated statistically significant antiviral effects, with adjusted *p*-values between *p* < 0.001 for coumarin and esculin and *p* < 0.0001 for esculetin and 4-methyl esculetin, indicating robust inhibition of viral activity compared with the positive control (PC). The corresponding confidence intervals (CIs) ranged for these compounds were coumarin (3,448,120 to 5,745,047), esculetin (4,337,745 to 6,634,672), esculin (3,898,953 to 6,195,880), and 4-methyl (4,253,941 to 6,631,487), all showing substantial mean differences in antiviral efficacy. However, umbelliferone (*p* = 0.058) and hesperetin (*p* = 0.0861) did not exhibit a significant antiviral effect, and neither did acetylated (*p* = 0.0649) or prenylated umbelliferones (*p* = 0.4014) (Figure 7D).

An inhibition assay using human bronchial epithelial (HBE) cells (Figure 7C) showed significant differences for coumarin and esculetin. Notably, encapsulated coumarin exhibited a strong inhibitory effect, with a mean difference of 19,333,333 (*p* = 0.0021), and esculetin showed a moderate effect (mean difference of 12,600,000, *p* = 0.0386). In contrast, esculin, 4-methyl, umbelliferone, and hesperetin demonstrated no significant inhibition compared with the PC, with *p*-values ranging from 0.1510 to 0.5758. The significance values vary between experiments with three (HBE) and eight (A549) concentration points because the number of data points directly influences statistical outcomes. A smaller number of points reduces the resolution and precision of the analysis, which likely contributes to the differences in significance observed in assays conducted on cultured A549 and HBE cells.

## 3. Discussion

The search for natural molecules with biopharmacological properties and easy extraction has always been a desire of the pharmaceutical industry, as well as a way of reducing the costs of producing these active compounds for final commercialization. Coumarins have been studied for a long time due to their biological properties, which include antifibrotic, anticoagulant, antimicrobial, and antiviral effects. This is primarily because the chemical basis of these compounds was used in the development of several currently available drugs, including Acenocoumarol, Dicumarol, Ticumarol, and Novobiocin [54].

The result of the cytotoxicity tests in this study shows low cell death and damage at high concentrations of coumarin, esculin, and umbelliferone, except for 4 methyl esculetin and esculetin, in which the toxic concentration was approximately four and three times higher, respectively. The HEp-2 cells were more resistant than A549 cells when it came to cell viability because, at the same concentrations of the compounds, A549 cells suffered a drop of around 20% in cell death. However, the concentrations are directly related to the inhibitory action of each of the coumarins. In the viral activity assays using rgRSV, viral inhibition was observed in the presence of esculetin and 4-methylcoumarin, molecules that, despite exhibiting slightly higher toxicity compared with other coumarins, demonstrated antiviral activity. In the fluorescence microscopy images, the reduction in the number of infected cells indicates that the viral infection was significantly inhibited.

Although assays using viruses tagged with green fluorescent protein offer valuable insights and are highly useful for result interpretation, they, like any method, have certain limitations. To further confirm the inhibitory effects of these compounds, a recombinant virus expressing the luciferase enzyme was employed. As the production of light from luciferase enzymatic activity is highly sensitive, even small amounts of virus infection can be detected with high precision. As it replicates, the virus produces the luciferase mRNA, which is translated into the luciferase protein. This system allows viral infections to be quantified at very early stages or in samples with low viral loads, where other methods, such as the detection of viral proteins, would be less sensitive. In addition, luciferase assays are very reproducible, and the amount of light generated is directly proportional to the quantity of virus replication in the cells.

In the antiviral action tests on A549 cells infected with RSV-luc, esculetin demonstrated approximately 50% inhibition at 24 h, which increased to 70% at 48 h; in contrast, 4-methyl esculetin inhibited the virus by about 40% at 24 h, with the inhibition doubling at 48 h. These results corroborate the inhibition tests using rgRSV. These tests also included the flavonoid hesperetin, a molecule that has been tested in vitro in interaction experiments with the N-terminal domain of the RSV nucleoprotein. These results were previously published by our group [19]. Hesperetin showed 57% viral inhibition in 24 h, but this inhibition was not maintained at 48 h.

The phenolic compounds tested are hydrophobic, so there was some doubt as to whether these molecules could permeate the cell membrane and reach the cytoplasm where virus replication takes place. In addition, their solubilization needs to be carried out in solvents such as DMSO, which are considered toxic to cells at certain concentrations. Therefore, to eliminate this type of toxicity and increase the solubility of the coumarins and hesperetin, we encapsulated them with the oligosaccharide β-cyclodextrin.

Our main interest was to analyze the antiviral potential of these coumarins and flavonoids in cultures of primary human respiratory tract epithelial cells, the natural target for RSV, which would be more reliable for what occurs in vivo. We used fully differentiated primary HBE cell cultures, since this cell type reproduces stimuli and presents barriers, such as cilia and mucus, as in the human respiratory system. In the antiviral tests on HBE cultures, the encapsulated molecules coumarin, esculetin, and hesperetin showed inhibition against RSV of 25, 15, and 7.5%, respectively, where coumarin revealed the most inhibition. Coumarin and hesperetin exhibited constant inhibition at increasing concentrations, which may have been due to saturation of the viral targets; that is, viral inhibition may have already reached a plateau, beyond which increasing the concentration of the drug does not result in further inhibition. Saturation occurs when the viral targets, proteins or receptors, are already completely occupied or blocked by the drug, and increasing the concentration does not increase efficacy [55]. Negative viral feedback represents another potential mechanism. In certain circumstances, a viral infection triggers an autoregulatory mechanism whereby the virus reduces its reproduction to adjust to the hostile environment the drug creates. This adaptive response can be triggered by the drug by blocking initial viral replication, which leads to continuous levels of viral inhibition independent of the higher antiviral concentrations [56]. There are also many viruses that have a functional reserve of enzymes or proteins that allow them to continue replicating even under partial inhibition of drugs. However, if the drug can inhibit viral replication over a certain limit, the capacity of the virus to compensate for this inhibition is reduced, and increasing the concentration of the drug has no additional effect. This can be seen with drugs that have a limiting antiviral effect, where the extra concentration of the drug does not result in further inhibition over the baseline level [57].

A different factor to consider is, for example, pharmacokinetic distribution. Drugs with a high affinity for specific tissues or cells can reach concentrations sufficient to inhibit the virus at a very low dose, and increasing the dose may not significantly change the concentration in the target viral compartments. This happens with intracellular antivirals that are rapidly concentrated in infected cells, causing rapid saturation of the active sites inside the cells [58]. Lastly, another assumption would be the high binding affinity between the drug and its viral target, explaining the constant inhibition at different concentrations. In some cases, where the drug has a low dissociation value (Kd), even low concentrations of the drug can result in almost complete occupation of the targets, leading to robust inhibition. If the drug binds strongly to the viral target, increasing the concentration may not provide additional advantages, since the majority of the targets are already saturated [59].

According to the data analyzed to evaluate the balance between efficacy and toxicity, 4-methyl esculetin showed the highest Selectivity Index (SI = 14.78) than the other compounds tested, a molecule with excellent inhibitory potential and molecular scaffold suitable for further studies. The SI of esculetin (7.56) and hesperetin (7.39) indicates moderate antiviral activity, information that corroborates the Abdelmohsen et al. 2021 study, in which hesperetin has the potential to inhibit viral replication in the pathways modulating oxidative stress and inflammation in influenza A and SARS-CoV-2 [60]. The results obtained for umbelliferone (SI = 0.46) compared with the other compounds indicate that structural modifications to the structure of coumarins are crucial to increasing antiviral activity, an observation consistent with the results of Garg and collaborators, who reported that unmodified coumarins had lower antiviral potency against viruses such as chikungunya and dengue [61]. Also, Cheng and collaborators show that umbelliferone potentially suppresses the cellular entry of SARS-CoV-2 by preventing the protein binding activity of the S protein to ACE2, based on enzymatic assays and molecular docking analyses [62].

In statistical terms, the 4-methyl esculetin has a superior SI compared with hesperetin, and other coumarins were confirmed with Dunnett’s multiple comparisons test, demonstrating statistically significant differences (*p* < 0.05) between the groups when compared with the control. The 4-methyl esculetin showed a marked reduction in cytotoxicity (CC_50_ = 1.04 mM) with a strong inhibition of viral replication (IC_50_ = 0.07 mM). In contrast, hesperetin exhibited a higher cytotoxicity (CC_50_ = 2.67 mM) and a less potent inhibition (IC50 = 0.36 mM), with an SI of 7.39, consistent with its reported moderate antiviral effects in studies on influenza and SARS-CoV-2 [60]. In HBE cells, data demonstrate that the viral inhibition efficacy of coumarin and esculetin significantly modulated their cytotoxic profiles; these results emphasize that coumarin and esculetin show promising antiviral effects in HBE cells, making them better candidates for further preclinical testing, but an optimization that reduces the cytotoxicity of these coumarins is necessary.

Analyzing the data obtained in our study, we can infer that beta cyclodextrin worked as a vehicle for the compounds to be delivered from the cell cytoplasm and decrease the viral titer, but it was not yet possible in this study to find the stoichiometry between the B-cyclodextrin/compound complexes and determine which cellular or viral target these molecules are targeting. Our ongoing research aims to elucidate these critical parameters and confirm the mechanism of action of these promising antiviral agents, as well as the stoichiometry of the complexes. Our findings also contribute to the growing research on coumarin-based antiviral therapies, demonstrating their potent antiviral activity against RSV and optimizing the pharmacological properties of these coumarin derivatives. They can serve as powerful structures for the design of more effective and safer agents specific for RSV infections.

## 4. Materials and Methods

### 4.1. Polyphenolic Compounds

The coumarin (1,2 Benzopyrone) (catalog number: C4261), esculetin (6,7-Dihydroxycoumarin) (catalog number: Y0001611), esculin (6,7-Dihydroxycoumarin-6-glucoside) (catalog number: Y0001612), 4-methyl esculetin (5,7-Dihydroxycoumarin-4-methylcoumarin) (catalog number: 543594), umbelliferone (7-Hydroxycoumarin) (catalog number: H24003), and hesperetin (3′,5,7-Trihydroxy-4′-methoxyflavanone) (catalog number: H4125) were purchased from Sigma-Aldrich (St. Louis, MO, USA). A stock solution was prepared in DMSO to obtain stock solutions at 150 mM concentrations that were further dissolved in DMEM. Final DMSO concentration was less than 0.01%. The acetylated and prenylated umbelliferones were chemically modified from commercial umbelliferone, and the steps are described in Section 4.2.

### 4.2. General Chemical Procedures for the Synthesis of Modified Umbelliferones

Reagents, solvents, and deuterated chloroform were obtained from Merck^®^ (Rahway, NJ, USA). Thin-layer chromatography (TLC) plates, made of silica gel (8.0–12.0 µm, 200 µm), were sourced from Supelco^®^ (St. Louis, MO, USA). The plates were inspected under UV light at 254 nm and 365 nm and developed using anisaldehyde–sulfuric acid solution to monitor the progress of the reactions. The 1D-1H spectra were recorded at 25 °C on an NMR Bruker Avance III spectrometer of 14.1 T operating at a 1H frequency of 600 MHz (Bruker BioSpin GmbH, Ettlingen, Germany) and equipped with a cryogenically cooled Z-gradient probe. Tuning and matching procedures for equipment tuning, D2O frequency locking, and one-dimensional gradient shimming to ensure the homogeneity of the magnetic field were automatically carried out.

#### 4.2.1. Synthesis and Identification of Acetylated and Prenylated Umbelliferone

The modifications were performed following well-established acetylation and prenylation protocols from the literature, with slight adjustments. The resulting structures were confirmed by analyzing all relevant ^1^H NMR parameters, including chemical shifts (δH), integration, multiplicities (s = singlet; d = doublet; m = multiplet), and coupling constants (J in Hz). The data obtained were consistent with the expected structure and previously reported values in the literature [63,64].

##### Synthesis of 7-Acetoxycoumarin

The acetic anhydride (2.0 mL, 21.15 mmol) was slowly added to a stirred solution of umbelliferone (0.025 g, 0.15 mmol) in pyridine (2.0 mL). The mixture was stirred for 24 h at room temperature. The crude was concentrated under reduced pressure, acidified with aqueous solution of hydrochloric acid 10% (*v*/*v*, 15 mL), and extracted with dichloromethane (3 × 15 mL) to remove the remaining pyridine. The concentrated crude was purified through crystallization using methanol, producing 7-acetoxycumarin as a white crystal in a 76% yield (0.024 g); 1H NMR (600 MHz, CDCl3) δ 7.70 (d, J = 9.5 Hz, 1H, H-4), 7.49 (d, J = 8.4 Hz, 1H, H-5), 7.12 (d, J = 2.1 Hz, 1H, H-8), 7.06 (dd, J = 8.4, 2.2 Hz, 1H, H-6), 6.40 (d, J = 9.6 Hz, 1H, H-3), 2.34 (s, 3H, H-2′) (Appendix A) [63].

##### Synthesis of 7-Isopentenyloxycoumarin

The 3,3-Dimethylallyl bromide (0.042 mL, 0.37 mmol) was slowly added to a stirred solution of umbelliferone (0.050 g, 0.31 mmol) and potassium carbonate (0.085 g, 0.62 mmol) in acetone (3.0 mL). The mixture was stirred for 8 h at room temperature. Distilled water (20 mL) was added to the crude and extracted with ethyl acetate (3 × 20 mL). The organic layer was dried with anhydrous magnesium sulfate and concentrated under reduced pressure. The concentrated crude was purified through crystallization using methanol, producing 7-isopentenyloxycoumarin as a white crystal in an 86% yield (0.061 g); 1H NMR (600 MHz, CDCl3) δ 7.63 (d, J = 9.5 Hz, 1H, H-4), 7.36 (d, J = 8.6 Hz, 1H, H-5), 6.85 (dd, J = 8.6, 2.4 Hz, 1H, H-6), 6.82 (d, J = 2.4 Hz, 1H, H-8), 6.24 (d, J = 9.4 Hz, 1H, H-3), 5.50–5.45 (m, 1H, H-2′), 4.58 (d, J = 6.7 Hz, 2H, H-1′), 1.81 (s, 3H, H-4′), 1.77 (s, 3H, H-5′) (Appendix A) [64].

### 4.3. Cell Culture

The HEp-2 and A549 cell lines were cultured in DMEM supplemented with 10% fetal bovine serum, 100 U/mL penicillin, 25 μg/mL streptomycin, 2 mM l-glutamine, and 0.01 M HEPES (complete DMEM) and incubated in humidified 5% CO_2_ at 37 °C. Primary well-differentiated human bronchial epithelial cultures (HBE) progenitors were isolated from donor airways, grown for a week to confluency, and slowly frozen for later use. As needed, progenitor cells were thawed and plated on 0.4 μM pore Transwells (Corning) membranes, 6.5 mm or 12 mm in diameter, fed with ALI medium supplemented with ROCK inhibitor in both the apical and basolateral chambers. Medium in both chambers was replaced with fresh medium every 2–3 days. At 7 days, when the cells were confluent and had formed tight junctions, as demonstrated by electrical resistance, the apical medium was removed, and the basal medium was replaced with complete Pneumacult-ALI Medium (STEMCELL Technologies) for 3 weeks until completely differentiated. The medium was replaced with fresh medium, and the apical surface was washed with 100 μL of DMEM before initiating inhibitor tests [65].

### 4.4. Cytotoxicity

The MTT assay (Protocol Merck-Sigma Aldrich) was used to quantify cell viability. The formazan crystals were dissolved in 0.001% DMSO, and the absorbance of the resulting solution was measured at 500–600 nanometers with a multiwell spectrophotometer [66,67,68]. For experiments, 5 × 104 HEp-2 or A549 cells were added to each well of a 96-well plate. After 24 h of incubation, cell monolayers were checked visually with an inverted microscope for the integrity of the monolayer. Wells with 60% to 80% cell confluence were selected for the experiment. The tested compounds [coumarin, esculetin, esculin, 4-methyl esculetin, umbelliferone, modified umbelliferones (acetylated and prenylated), and hesperetin] were weighed and dissolved in DMSO to prepare 150 mM stocks. Compounds were serially diluted 2-fold in cell culture medium in 96-well round bottom plates with three replicate wells for each tested concentration. In the HEp-2 cell experiment, 500 µM was the highest concentration used.

In A549 cells, the starting concentration varied from 500 to 75 µM, depending on findings in preliminary experiments, followed by six 1:2 dilutions. Cytotoxicity experiments were also carried out with comparable serial dilutions of DMSO, the solvent in which the compounds were diluted. The MTT working solution (2.5 mg/mL) was prepared in DMEM medium. The culture medium for each well was replaced with 0.1 mL of the MTT solution and incubated at 37 °C in 5% CO_2_ for 3 h. The solution was then aspirated from the wells, and 100 µL of DMSO (99%) was added. After 15 min, the optical density was measured at a wavelength of 560 nm, plotted, and the CC50 (concentration of each test material that reduces cell viability by 50%) was calculated. As a negative control for cell viability, cells were treated with 1 μM staurosporine. Staurosporine is an alkaloid isolated from the culture broth of Streptomyces staurosporeus that is a potent kinase inhibitor inducing the apoptosis of cells [69,70]. Compounds were added to the medium before and after inoculation. All cytotoxicity tests were performed in triplicate.

### 4.5. Antiviral Activity

The antiviral activity of test compounds against RSV was assessed for 2- and 5-fold dilutions of the compounds, starting from a concentration of the compound that supported >78% cell viability when incubated with HEp-2 cell monolayer (for coumarin, esculin, umbelliferone and hesperetin 500 μM, esculetin 250 μM and 4-methyl 125 μM), A549 cell monolayer and HBE culture (for coumarin, esculin, umbelliferone and hesperetin 300 μM, esculetin 125 μM and 4-methyl 75 μM) under normal culture conditions (37 °C and 5% CO_2_) for 2 to 3 days. Cell monolayers were inoculated with RSV (M.O.I 0.02) for 2 h, after which the virus was removed, and the cells were washed twice with fresh medium and replaced with medium containing the compounds at the desired concentrations and incubated at 37 °C and 5% CO_2_ for 2 to 3 days. The compounds were added before and after infection to maintain the chemical balance of the molecules in the cellular environment. All inhibition tests were performed in triplicate.

The recombinant RSV strains used expressed green fluorescent protein (GFP) or Renilla luciferase. Both viruses were based on the A2 strain [71], one expressing Green Fluorescence Protein (GFP), rgRSV [72] and the other expressing Renilla luciferase (rA2-Rluc) [73]. Both were used to inoculate A549 cells and HBE cultures at an M.O.I. 0.02. Both RSV stocks were produced in HEp-2 cells in DMEM-10% FBS (fetal bovine serum). GFP-expressing viruses were titrated on HEp-2 cells by inoculating for 2 h at 37 °C at an M.O.I. 0.02 and counting GFP foci at 24/48/72 hpi. To quantify luciferase production, A549 cell monolayers or HBE cultures were inoculated at 37 °C and incubated in 5% CO_2_ for 2 h, washed, and fresh medium added. After 48 h, the cells were washed once with PBS, then a lysis buffer was added to the cells, and they were shaken for 15 min. Then, the lysed cells were transferred to a black 96-well plate, and luciferase substrate was added. Luminescence was measured using a Multi+ luminometer (Promega, Madison, WI, USA). Viruses and cells were determined to be mycoplasma-free using the ATCC Mycoplasma detection kit. The viral growth curves were created with GraphPad Prism 9.0.2 following calculation:$$\mathrm{Viral}\;\mathrm{inhibition}\;\left(\%\right)=\left(1-\frac{\mathrm{Activity}\;\mathrm{of}\;\mathrm{the}\;\mathrm{Treated}\;\mathrm{Sample}}{\mathrm{Activity}\;\mathrm{of}\;\mathrm{the}\;\mathrm{Control}\;\mathrm{Sample}}\right)\times 100$$

### 4.6. Encapsulation of the Compounds

Compounds were encapsulated using a 1 to 2 ratio of β-cyclodextrin, which was diluted in 10 mL of milliQ water. This solution was placed in a 60 °C water bath for 16 h and lyophilized. The resulting powder was washed with ethanol to remove the compounds that had not been encapsulated. The ethanol wash solution was measured in the UV-spectrometer to determine the amount of compound that had not been encapsulated [74,75,76].

#### Nuclear Magnetic Resonance (NMR)

The 1D-1H spectra were recorded at 25 °C on an NMR Bruker Avance III spectrometer of 14.1 T operating at a 1H frequency of 600 MHz (Bruker BioSpin GmbH, Germany) and equipped with a cryogenically cooled Z-gradient probe. Tuning and matching procedures for equipment tuning, D2O frequency locking, and one-dimensional gradient shimming to ensure the homogeneity of the magnetic field were automatically carried out.

### 4.7. Statistical Analysis

Data analyses and visualizations were conducted using GraphPad Prism software (version 10.1.0; GraphPad Software, La Jolla, CA, USA). Statistical significance was evaluated using two-way ANOVA followed by Dunnett’s multiple comparisons test. Adjusted *p*-values were calculated to account for multiple comparisons, and differences were considered statistically significant at *p* < 0.05 *. Significant results include *p* < 0.01 for **, *p* < 0.001 for ***, and *p* < 0.0001 for ****.

### 4.8. Selectivity Index (SI)

The concentration of 50% cytotoxicity (CC_50_) was estimated for all compounds by linear interpolation from the concentrations corresponding to 90% (CC_90_) and 100% (CC_100_) cytotoxicity. The CC_50_ value was calculated using the following equation:$${\mathrm{CC}}_{50}={\mathrm{CC}}_{90}-\left.\left(\;\frac{\left({\mathrm{CC}}_{90}-{\mathrm{CC}}_{100}\right)\times \left(90-50\right)}{\left(90-100\right)}\right.\right)$$
where CC_90_ and CC_100_ represent the concentrations at which 90% and 100% cytotoxicity were observed, respectively. The LINEST function in Microsoft Excel was utilized to perform linear regression. For estimating the IC_50_ values not determined experimentally for coumarin, esculin, and umbelliferone, interpolation methods were also used. Selectivity Index (SI) was calculated by dividing the IC_50_ of the antiviral compound by the CC_50_ of the corresponding cytotoxicity data. The SI was calculated for A549 cells.
$$\mathrm{SI}=\frac{{\mathrm{IC}}_{50}(\mathrm{antiviral})}{{\mathrm{CC}}_{50}(\mathrm{cytotoxicity})}$$

## 5. Conclusions

Coumarins inhibit RSV replication in cell culture and even manage to overcome the mucus barriers of the HBE cultures. In our study, β-cyclodextrin was essential for some of the coumarins to enter the cell and therefore to reach their targets, corroborating the literature that it is an excellent vehicle for transporting small hydrophobic molecules. However, more research is required to determine the optimal stoichiometry of the β-cyclodextrin/coumarin and hesperetin complexes, as well as to identify their viral or cellular targets in relation to RSV inhibition. Furthermore, more studies are required to propose chemical modifications to the chemical structure of these phenolic compounds in order to increase their anti-RSV action and decrease their cytotoxicity.

## Figures and Tables

**Figure 1 ijms-25-13301-f001:**
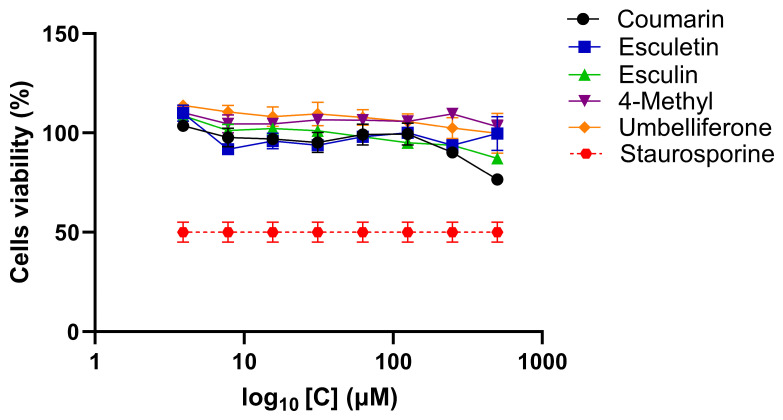
Cytotoxicity assay by MTT of natural coumarins on HEp-2 cells following 48 h of incubation with the test compounds coumarin, esculetin, esculin, 4-methyl esculetin, and umbelliferone. The hexagon with dotted red line represents Staurosporine (1 mM), negative control, in which 50% of the cells remain viable.

**Figure 2 ijms-25-13301-f002:**
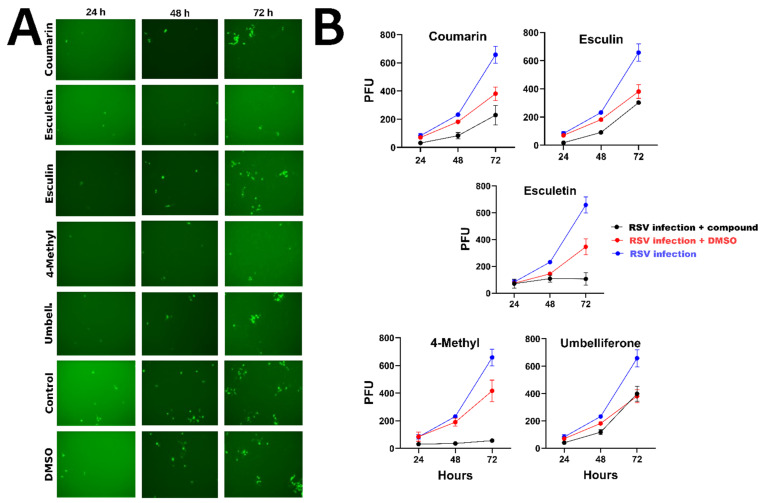
(**A**) Fluorescence microscopy images taken using the green fluorescent protein (GFP) filter at 24, 48, and 72 h to demonstrate the spread of virus infection in treated cells compared with the untreated control or the DMSO-treated control. Treatment of rgRSV infection of HEp-2 cells with coumarin, esculetin, esculin, 4-methyl esculetin, and umbelliferone (before and after infection). HEp-2 cells were cultured overnight and inoculated with rgRSV. (**B**) The black line represents RSV infection in the presence of the coumarins, the red line is the DMSO solvent control, and the blue line represents the infection without compounds.

**Figure 3 ijms-25-13301-f003:**
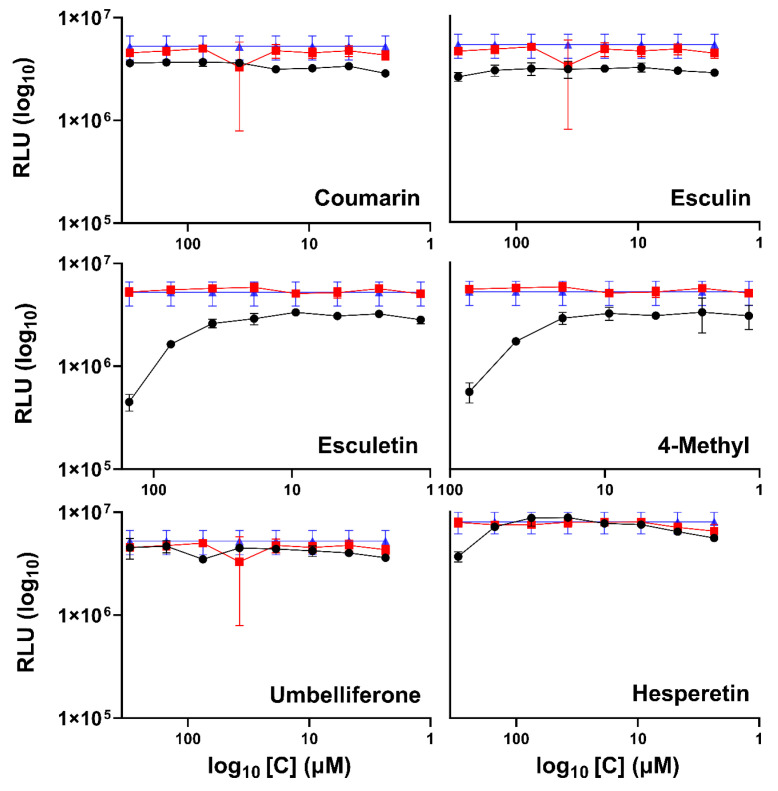
Inhibition assay with RSV-luc on A549 cells at 48 h for coumarins and 24 h for hesperetin at. The black lines and circles represent RSV infection in the presence of the compounds (coumarin, esculetin, esculin, 4-methyl esculetin, umbelliferone and hesperetin), red lines and squares represent RSV infection in the presence of the DMSO, and blue lines and triangle represents RSV infection in the absence of a compound.

**Figure 4 ijms-25-13301-f004:**
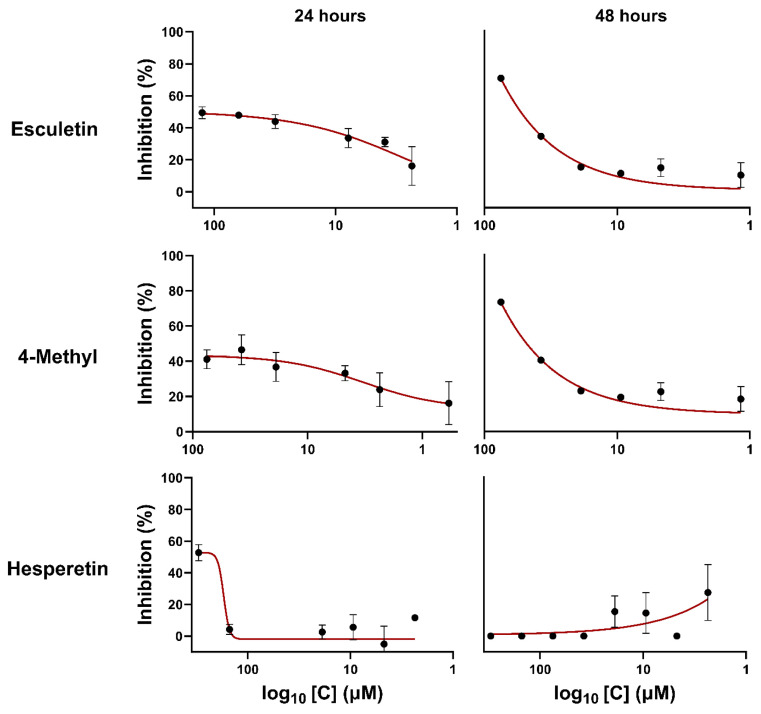
Inhibition plot of RSV-luc infection of A549 in the presence of compounds esculetin, 4-methyl esculetin, and hesperetin, at 24 and 48 h.

**Figure 5 ijms-25-13301-f005:**
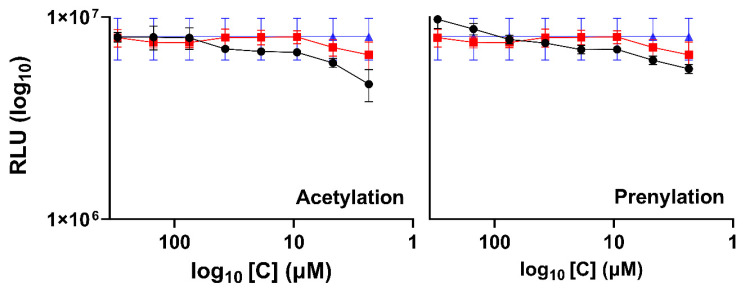
Inhibition assay on A549 cell with RSV-luc for acetylated and prenylated umbelliferone at 24 h. The black lines and circles represent RSV infection in the presence of the compounds (coumarin, esculetin, esculin, 4-methyl esculetin, umbelliferone and hesperetin), red lines and squares represent RSV infection in the presence of the DMSO (solvent), and blue lines and triangles represent RSV infection without DMSO or compounds.

**Figure 6 ijms-25-13301-f006:**
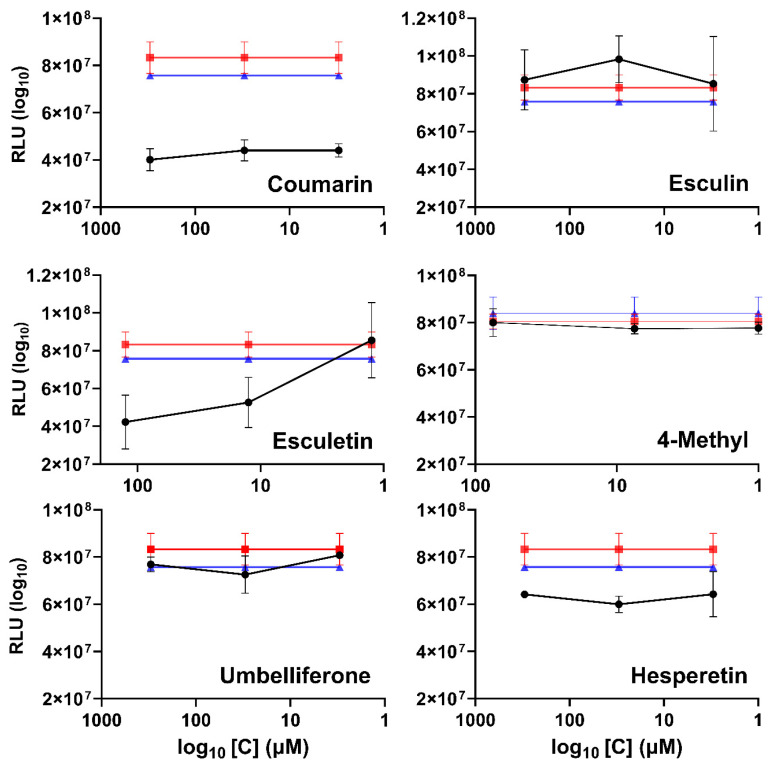
Inhibition assay of the encapsulated compounds on differentiated HBE cultures inoculated with RSV-luc, at 48 h. The black line and circles represents RSV infection in the presence of the encapsulated compounds (coumarin, esculetin, esculin, 4-methyl esculetin, umbelliferone and hesperetin), the red line and squares shows the RSV infection in the presence only of β-cyclodextrin, and the blue line and triangle represents the RSV infection with no additions.

**Figure 7 ijms-25-13301-f007:**
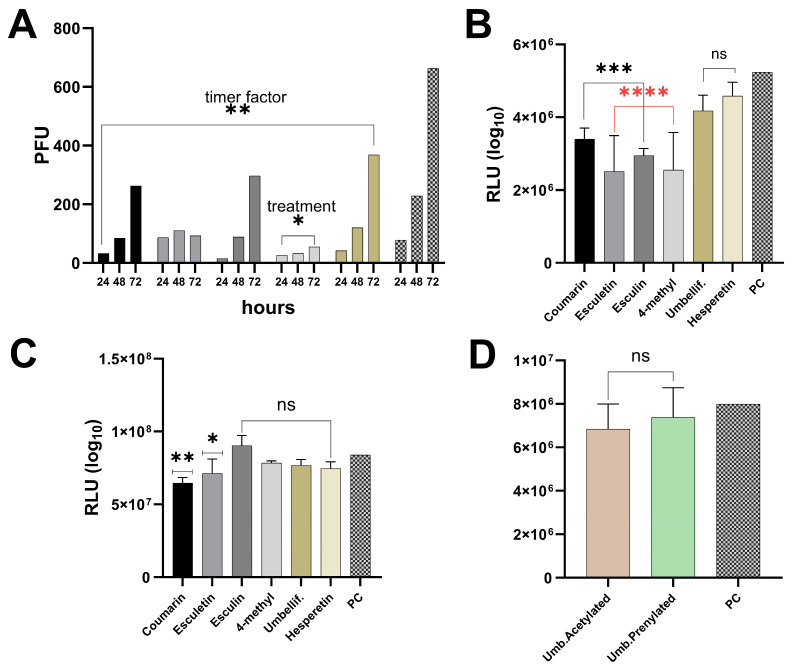
RSV replication and spread in different cell lines and in primary HBE cultures in the presence of coumarins, hesperetin and modified umbelliferones. GFP-expressing RSV replication and spread in (**A**) HEp-2 cells as measured by virus yield in focus forming units ± standard deviation (SD). Replication and spread of RSV expressing luciferase, measured in relative light units (RLU) ± SD in A549 cells (**B**,**D**), and (**C**) HBE cultures. Statistical significance Significant differences are indicated as follows: * *p* < 0.05, ** *p* < 0.01, *** *p* < 0.001, and **** *p* < 0.0001 and ns is non- significant.

**Table 1 ijms-25-13301-t001:** Cytotoxicity and antiviral activity of coumarins and hesperetin. The table lists the CC_50_ (concentration causing 50% cytotoxicity), IC_50_ (concentration inhibiting 50% viral replication), and the corresponding Selectivity Index (SI) for each compound.

Compounds	CC_50_ (mM)	IC_50_ (mM)	SI
4-methyl esculetin	1.04	0.07	14.78
Esculetin	1.41	0.18	7.56
Hesperetin	2.67	0.36	7.39
Coumarin	3.56	1.65	2.16
Esculin	1.93	1.59	1.21
Umbelliferone	2.40	5.14	0.46

## Data Availability

The original contributions presented in the study are included in the article/Appendix A, further inquiries can be directed to the corresponding author/s.

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
