# Peer review of "Coumarins and Hesperetin Inhibit Human Respiratory Syncytial Virus Infection"

_ijms, 2024, doi:10.3390/ijms252413301_

Round 1
Reviewer 1 Report
Comments and Suggestions for Authors
After a thorough review of the manuscript entitled “Coumarins and Hesperetin Inhibit Human Respiratory Syncytial Virus Infection”, I highlighted several points that should be taken into consideration by the authors to improve the study. Authors should make substantial changes to the manuscript and include statistical analysis of the results.
1. Lines 73-79: In the introduction, I suggest that the authors include more information about the antiviral potential of hesperetin. This flavonoid has been well reported in the literature regarding its anti-SARS-CoV-2 effect.
Agrawal, P. K., Agrawal, C., & Blunden, G. (2021). Pharmacological significance of hesperidin and hesperetin, two citrus flavonoids, as promising antiviral compounds for prophylaxis against and combating COVID-19. Natural Product Communications, 16(10), 1934578X211042540.
Khezri, M. R., Ghasemnejad‐Berenji, M., & Moloodsouri, D. (2022). Hesperetin and the PI3K/AKT pathway: Could their interaction play a role in the entry and replication of the SARS‐CoV‐2?. Journal of Food Biochemistry, 46(9), e14212.
Zalpoor, H., Bakhtiyari, M., Shapourian, H., Rostampour, P., Tavakol, C., & Nabi-Afjadi, M. (2022). Hesperetin as an anti-SARS-CoV-2 agent can inhibit COVID-19-associated cancer progression by suppressing intracellular signaling pathways. Inflammopharmacology, 30(5), 1533-1539.
2. Results section. Figures 1, 2, 3, 4, 5 and 6: It is necessary to apply a statistical test to all experiments performed. The graphs presented in the results section do not show whether there was a significant difference between the different treatments (compounds and control). It is recommended to apply analysis of variance (ANOVA) and Tukey's test.
3. Lines 209-219: It is important to include citations to support the discussion of the results of this study.
4. Overall, the discussion text is very comprehensive. Authors should discuss the results critically and based on other studies that evaluated the antiviral potential of coumarins and flavonoids against respiratory syncytial virus.
5. Line 293: Was the cytotoxicity test performed in triplicate? This information is not included in the text.
6. Subsection 4.3.: Please provide information about the controls used in the antiviral activity.
Why was the broad-spectrum antiviral drug (ribavirin) indicated to treat respiratory syncytial virus infections not used as a positive control in this study? It would be interesting to compare the results of the commercial drug in relation to natural products.
7. Line 319: Was antiviral activity performed in triplicate? This information is not included in the text.
8. Lines 320-322: It is necessary to specify which compound supported >80% cell viability.
9. Lines 326-328: Authors should include information on the specific concentrations of each compound used in the antiviral assay.
10. Lines 360-399: I didn't understand this information. Have all chemical compounds been synthesized? Why didn't the authors report the synthesis process for all of them? Please provide more details to improve understanding.
Typically, this information about chemicals and general procedures is added in the first subsection of the methodology.
11. I did not find information about the process of obtaining hesperetin in the methodology of this manuscript. If any phytochemical used in this study was obtained commercially, I suggest specifying and citing the name of the company. Additionally, it is necessary to include information about the batch number and purity of the compounds.
12. At the end of the materials and methods section, authors should include a subsection titled “Statistical analysis.” It is necessary to inform which statistical tests were used to analyze the results. The software should also be mentioned.
Author Response
Coumarins and Hesperetin Inhibit Human Respiratory Syncytial Virus Infection
Jéssica Maróstica de Sá, Ilada Thongpan, Jefferson de Souza Busso, Thaina da Silva Rodrigues, Phylip Chen, Alvaro Luiz Helena, Luis Octavio Regasini, Marcelo Andres Fossey, Ícaro Putinhon Caruso, Fátima Pereira de Souza, Mark E. Peeples
Email: fatima.p.souza@unesp.br
December 02, 2024
Response letter to the reviewers.
We sincerely thank you for the suggestions regarding potential reviewers for the evaluation of our manuscript submitted to the journal. These recommendations are of great importance in ensuring a critical and constructive review of the work, contributing to the refinement of the research and upholding the scientific rigor of the article. We deeply appreciate the care and attention devoted to this step of the editorial process and extend our appreciation for the professionalism and consideration demonstrated.
Reviwer 1
1. Lines 73-79: In the introduction, I suggest that the authors include more information about the antiviral potential of hesperetin. This flavonoid has been well reported in the literature regarding its anti-SARS-CoV-2 effect.
We added information in the introduction about hesperetin include your reference suggestions (lines 76-96)
2. Results section. Figures 1, 2, 3, 4, 5 and 6: It is necessary to apply a statistical test to all experiments performed. The graphs presented in the results section do not show whether there was a significant difference between the different treatments (compounds and control). It is recommended to apply analysis of variance (ANOVA) and Tukey's test.
We performed statistical tests to analyze the antiviral efficacy of the molecules studied using ANOVA and Dunnett’s multiple comparisons test, which can compare the molecules with the control. Based on the data, we created graphs which are shown in figure 7 and the results with the parameters obtained (206-245), as well as adding the results to the discussion (367-378).
3. Lines 209-219: It is important to include citations to support the discussion of the results of this study.
The citation was added (line 208).
4. Overall, the discussion text is very comprehensive. Authors should discuss the results critically and based on other studies that evaluated the antiviral potential of coumarins and flavonoids against respiratory syncytial virus.
Studies that discuss the antiviral action of coumarins and flavonoids for viruses such as SARS, among others, were added to the discussion as comparisons, since there is no study focused on RSV (lines 353-378)
5. Line 293: Was the cytotoxicity test performed in triplicate? This information is not included in the text.
The cytotoxicity assay was performed in triplicate. This information was added (line 495)
6. Subsection 4.3.: Please provide information about the controls used in the antiviral activity.
Why was the broad-spectrum antiviral drug (ribavirin) indicated to treat respiratory syncytial virus infections not used as a positive control in this study? It would be interesting to compare the results of the commercial drug in relation to natural products.
Thank you once again for your contributions to our research. We would also like to explain why we did not use ribavirin as an inhibition control. We chose not to use ribavirin in the experiments for reasons related to its non-specificity against RSV, since ribavirin is a broad-spectrum antiviral that acts mainly by inhibiting viral RNA synthesis through generic mechanisms. Despite its in vitro efficacy against RSV in some experimental conditions, its non-specificity and broad action on different RNA viruses make it difficult to accurately assess its specific effects against RSV. As our study focused on investigating the specific antiviral potential against RSV, we used suitable negative and infected controls to validate our results, such as cells above 80% viability, which provides us with certainty regarding cell death and the effectiveness of inhibition by the compounds. We therefore felt that the inclusion of ribavirin as a control would not be essential for the objectives of the work. As we will continue the research with the coumarins that have shown the best results, we may include ribavirin in future studies for comparison, as you suggested.
7. Line 319: Was antiviral activity performed in triplicate? This information is not included in the text.
The antiviral assay was performed in triplicate. This information was added (line 510).
8. Lines 320-322: It is necessary to specify which compound supported >80% cell viability.
The concentration of each compound used in the experiments, in which the percentage of cell viability was above 78%, has been added to the text (lines 500-504).
9. Lines 326-328: Authors should include information on the specific concentrations of each compound used in the antiviral assay.
This information is also included in lines 500-504.
10. Lines 360-399: I didn't understand this information. Have all chemical compounds been synthesized? Why didn't the authors report the synthesis process for all of them? Please provide more details to improve understanding.
Coumarin, esculetin, esculin, 4-methyl, umbelliferone and hesperetin were purchased. And the acetylated and prenylated umbelliferones were modified from the purchased umbelliferone. We added all information in the lines 391-402.
Typically, this information about chemicals and general procedures is added in the first subsection of the methodology.
We have changed the information on the modification for the second session in the material and methods, which is just below the information on the purchased compounds (lines 404- 450)
11. I did not find information about the process of obtaining hesperetin in the methodology of this manuscript. If any phytochemical used in this study was obtained commercially, I suggest specifying and citing the name of the company. Additionally, it is necessary to include information about the batch number and purity of the compounds.
Hesperetin is a purchase. This information was included in the material and methods in lines 397 and 398.
12. At the end of the materials and methods section, authors should include a subsection titled “Statistical analysis.” It is necessary to inform which statistical tests were used to analyze the results. The software should also be mentioned.
We added in the materials and methods a session for “Statistical analysis” (lines 546-553).

Reviewer 2 Report
Comments and Suggestions for Authors
Thank you for the opportunity to review the original manuscript entitled "Coumarins and Hesperetin Inhibit Human Respiratory Syncytial Virus Infection".
Here are my comments:
1. Figure 1 can be improved by changing bars into lines. CC50 is not determined within the concentration range used. Thus selectivity index calculation is not performed but should be provided. there is a grammar mistake in the figure caption "cells are had died" - line 115-116.
2. Minor English editing is needed to correct small language mistakes such as '"Our research group searching for antiviral agents of natural origin that can reduce or prevent RSV infection, as coumarins and flavonoids" - the verb is missing, and others.
Author Response
Coumarins and Hesperetin Inhibit Human Respiratory Syncytial Virus Infection
Jéssica Maróstica de Sá, Ilada Thongpan, Jefferson de Souza Busso, Thaina da Silva Rodrigues, Phylip Chen, Alvaro Luiz Helena, Luis Octavio Regasini, Marcelo Andres Fossey, Ícaro Putinhon Caruso, Fátima Pereira de Souza, Mark E. Peeples
Email: fatima.p.souza@unesp.br
December 02, 2024
Response letter to the reviewers.
We sincerely thank you for the suggestions regarding potential reviewers for the evaluation of our manuscript submitted to the journal. These recommendations are of great importance in ensuring a critical and constructive review of the work, contributing to the refinement of the research and upholding the scientific rigor of the article. We deeply appreciate the care and attention devoted to this step of the editorial process and extend our appreciation for the professionalism and consideration demonstrated.
Reviwer 2:
1. Figure 1 can be improved by changing bars into lines. CC50 is not determined within the concentration range used. Thus selectivity index calculation is not performed but should be provided. there is a grammar mistake in the figure caption "cells are had died" - line 115-116.
The graph has been replaced by the line graph (line 125). A table has also been created with the index selectivity for all the compounds (lines 248- 272), as well as a discussion of the results (lines 353-366). We have also added a section for index selectivity (lines 555-570) to the materials and methods. The grammar of the line has been corrected (line 130).
2. Minor English editing is needed to correct small language mistakes such as '"Our research group searching for antiviral agents of natural origin that can reduce or prevent RSV infection, as coumarins and flavonoids" - the verb is missing, and others.
The error in the sentence has been corrected (lines 15-17) and we have also revised it to correct other sentences.

Round 2
Reviewer 1 Report
Comments and Suggestions for Authors
The authors responded to my comments and necessary changes were made to the manuscript. However, I do not understand why the authors named subsection 2.6. of "Statistical analysis" in the results text. Please replace the title of this subsection.
The subsection 4.7. "Statistical analysis" included in the methodology is sufficient and describes the statistical tests and software used. There is no need to repeat this topic in the results.
In addition, information on the selectivity index (section 2.7, lines 256-279) should be incorporated in section 2.1. "Cytotoxicity assay of natural and modified coumarins and hesperetin".
Author Response
Coumarins and Hesperetin Inhibit Human Respiratory Syncytial Virus Infection
Jéssica Maróstica de Sá, Ilada Thongpan, Jefferson de Souza Busso, Thaina da Silva Rodrigues, Phylip Chen, Alvaro Luiz Helena, Luis Octavio Regasini, Marcelo Andres Fossey, Ícaro Putinhon Caruso, Fátima Pereira de Souza, Mark E. Peeples
Email: fatima.p.souza@unesp.br
December 02, 2024
We would like to thank you again for your suggestions, they were extremely important in enriching our manuscript.
Reviewer 1
The authors responded to my comments and necessary changes were made to the manuscript. However, I do not understand why the authors named subsection 2.6. of "Statistical analysis" in the results text. Please replace the title of this subsection.
We changed the title to “Comparative evaluation of antiviral effects” (line 233)
The subsection 4.7. "Statistical analysis" included in the methodology is sufficient and describes the statistical tests and software used. There is no need to repeat this topic in the results.
The sentence similar to materials and methods has been removed from the first paragraph of session 2.6.
In addition, information on the selectivity index (section 2.7, lines 256-279) should be incorporated in section 2.1. "Cytotoxicity assay of natural and modified coumarins and hesperetin".
The topic of index selectivity has been added to section 2.1 as subsection 2.1.1.
